# A Boosting Approach to Reinforcement Learning

**Nataly Brukhim**
Princeton University
nbrukhim@cs.princeton.edu

**Elad Hazan**
Princeton University
Google AI Princeton
ehazan@cs.princeton.edu

**Karan Singh**
Carnegie Mellon University
karansingh@cmu.edu

## Abstract

Reducing reinforcement learning to supervised learning is a well-studied and effective approach that leverages the benefits of compact function approximation to deal with large-scale Markov decision processes. Independently, the boosting methodology (e.g. AdaBoost) has proven to be indispensable in designing efficient and accurate classification algorithms by combining inaccurate *rules-of-thumb*.

In this paper, we take a further step: we reduce reinforcement learning to a sequence of weak learning problems. Since weak learners perform only marginally better than random guesses, such subroutines constitute a weaker assumption than the availability of an accurate supervised learning oracle. We prove that the sample complexity and running time bounds of the proposed method do not explicitly depend on the number of states.

While existing results on boosting operate on convex losses, the value function over policies is non-convex. We show how to use a non-convex variant of the Frank-Wolfe method for boosting, that additionally improves upon the known sample complexity and running time even for reductions to supervised learning.

## 1 Introduction

In reinforcement learning, Markov decision processes (MDP) model the mechanism of learning from rewards, as opposed to examples. Although the case of tabular MDPs is well understood, the main challenge in applying RL in the real-world is the size of the state space in practical domains.

This challenge of finding efficient and provable algorithms for MDPs with large state space is the focus of our study. Various techniques have been suggested and applied to cope with very large MDPs. One class of approaches attempts to approximate either the value or the transition function of the underlying MDP by using a parametric function class. Such approaches invariably make strong *realizability assumptions* to produce global optimality guarantees. Another class of approaches, *so-called* direct methods, produces a near-optimal policy that maximizes the expected return from a given policy class. To deal with the challenge of large (possibly innumerable) policy classes, a popular strategy [24] is to the frame policy search as a sequence of supervised learning problems. Such approaches yield global optimality guarantees under state coverage assumptions without reliance on realizability, and have inspired practical adaptations for sampling-based policy search.

In this paper, we study another methodology to derive provable algorithms for reinforcement learning: ensemble methods for aggregating weak or approximate algorithms into substantially more accurate solutions. Our proposal extends the methodology of boosting, typically used to solve supervised learning instances [32], to reinforcement learning. A typical boosting algorithm (e.g. AdaBoost)

36th Conference on Neural Information Processing Systems (NeurIPS 2022).

|  | Supervised weak learner | Online weak learner |
|---|---|---|
| Episodic model | $1/\alpha^4\varepsilon^5$ | $1/\alpha^2\varepsilon^3$ |
| Rollouts w. $\nu$-resets | $1/\alpha^4\varepsilon^6$ | $1/\alpha^2\varepsilon^4$ |

Table 1: Sample complexity of the proposed algorithms for different $\alpha$-weak learning models (supervised & online) and modes of accessing the MDP (rollouts & rollouts with reset distribution $\nu$), in terms of $\epsilon$ and $\alpha$, suppressing other terms. This work is the first to introduce a reduction of RL to *weak* supervised learning. See Theorem 7 for details.

|  | Supervised strong learner | |
|---|---|---|
|  | This work (Corollary 8) | CPI [24] |
| Episodic model | $1/\varepsilon^3$ | $1/\varepsilon^4$ |
| Rollouts w. $\nu$-resets | $1/\varepsilon^4$ | $1/\varepsilon^4$ |

Table 2: Compared to previous work [24], the table shows sample complexity of the proposed algorithm for a strong ($\alpha = 1$) supervised learning model and different modes of accessing the MDP.

iteratively constructs a near-optimal classifier by combining computationally cheap, yet inaccurate *rules-of-thumb*. Unlike RL reductions to supervised learning which assume the existence of an efficient and accurate classification or regression procedure, the proposed algorithms builds on learning algorithms that perform only ever-so-slightly better than a random guess, and which thus may be produced cheaply both in computational and statistical terms.

Concretely, we assume access to a weak learner: an efficient sample-based procedure that is capable of generating an approximate solution to any weighted multi-class objective over a fixed policy class. We describe an algorithm that iteratively calls this procedure on carefully constructed new objectives, and aggregates the solution into a single policy. We prove that after sufficiently many iterations, our resulting policy has competitive global gurantees on performacnce. Interestingly, unlike boosting algorithms for regression and classification, our resulting aggregation of weak learners is non-linear.

### 1.1 Challenges and techniques

Reinforcement learning is quite different from supervised learning and several difficulties have to be circumvented for boosting to work. Among the challenges that the reinforcement learning setting presents, consider the following,

(a) The value function is not a convex or concave function of the policy. This is true even in the tabular case, and even more so if we use a parameterized policy class.

(b) The transition matrix is unknown, or prohibitively large to manipulate for large state spaces. This means that even evaluation of a policy cannot be exact, and can only be computed approximately.

(c) It is unrealistic to expect a weak learner that attains near-optimal value for a given linear objective over the policy class. At most one can hope for a multiplicative and/or additive approximation of the overall value.

Our approach overcomes these challenges by applied several new as well as recently developed techniques. To overcome the nonconvexity of the value function, we use a novel variant of the Frank-Wolfe optimization algorithm that simultaneously delivers on two guarantees. First, it finds a first order stationary point with near-optimal rate. Secondly, if the objective happens to admit a certain gradient domination property, an important generalization of convexity, it also guarantees near optimal value. The application of the nonconvex Frank-Wolfe method is justified due to previous recent investigation of the policy gradient algorithm [2, 1], which identified conditions under which the value function is gradient dominated.

The second information-theoretic challenge of the unknown transition function is overcome by careful algorithmic design: our boosting algorithm requires only samples of the transitions and rewards, obtained by rollouts on the MDP.

The third challenge is perhaps the most difficult to overcome. Thus far, the use of the Frank-Wolfe method in reinforcement learning did not include a multiplicative approximation, which is critical for our application. We adapt the techniques used for boosting in online convex optimization [19] with a multiplicative weak learner to our setting, by non-linearly aggregating (using a 2-layer network) the weak learners. This aspect is perhaps of general interest to boosting algorithm design, which is mostly based on linear aggregation.

## 1.2 Our contributions

Our main contribution is a novel efficient boosting algorithm for reinforcement learning. Our techniques apply in various settings and the sample complexity bounds of all of our results are summarized in Tables 1 and 2.

The input to this algorithm is a weak learning method capable of approximately solving a weighted multi-class problem instance over a certain policy class. The output of the algorithm is a policy which does not belong to the original class considered, hence being an instance of *improper* learning. It is rather a non-linear aggregation of policies from the original class, according to a two-layer neural network. This is a result of the two-tier structure of our algorithm: an outer loop of non-convex Frank-Wolfe method, and an inner loop of online convex optimization based boosting. The final policy comes with provable global optimality guarantees.

Beyond novelty of techniques, an important contribution (Table 1) of our work is to highlight the quantitative difference in guarantees that depend on the mode of accessing the MDP (episodic rollouts vs. access to an exploratory reset distrbution) and the nature of the weak learners (online vs statistical), thus indicating that some algorithmic choices may be preferable compared to others in terms of speed of convergence and sample complexity.

As with existing reductions to supervised learning [24], these global convergence guarantees happen under appropriate state coverage assumptions either via access to a reset distribution that has some overlap with the state distribution of the optimal policy, or by constraining the policy class to policies that explore sufficiently. Yet another contribution of our work is to show an improved sample complexity result in the latter setting, *even when considering reductions to supervised learning instances*. This improvement in convergence in well-studied settings is documented in Table 2.

## 1.3 Related work

Reinforcement learning approaches for dealing with large-scale MDPs rely on function approximation [35]. Such function approximation may be performed on the underlying conditional probability of transition (e.g. [34, 21]) or the value function (e.g. [37, 36]). The provable guarantees in such methods come at the cost of strong realizability assumptions. In contrast, the so-called direct approaches attempt policy search over an appropriate policy class [2, 1], and rely on making making incremental updates, such as variants of Conservative Policy Iteration (CPI) [24, 33, 4], and Policy Search by Dynamic Programming (PSDP)[6]. These provide convergence guarantees under appropriate state coverage assumptions comparable to ones made in this work.

Our boosting approach for provable RL builds on the vast literature of boosting for supervised learning [32], and recently online learning [27, 12, 13, 7, 22, 23]. One of the crucial techniques important for our application is the extension of boosting to the online convex optimization setting, with bandit information [10], and critically with a multiplicative weak learner [19]. This latter technique implies a non-linear aggregation of the weak learners. Non-linear boosting was only recently investigated in the context of classification [5], where it was shown to potentially enable significantly more efficient boosting. Another work on boosting in the context of control of dynamical systems [3]. However, this work critically requires knowledge of the underlying dynamics (transitions) and makes convexity assumptions, which we do not, and cannot cope with a multiplicative approximate weak learner.

The Frank-Wolfe algorithm is extensively used in machine learning, see e.g. [20], references therein, and recent progress in stochastic Frank-Wolfe methods [16, 28, 11, 39]. Recent literature has applied a variant of this algorithm to reinforcement learning in the context of state space exploration [18].

## 2  Preliminaries

**Optimization.**  We say that a differentiable function $f : \mathcal{K} \mapsto \mathbb{R}$ over some domain $\mathcal{K} \subset \mathbb{R}^d$ is $L$-smooth with respect to some norm $\| \cdot \|_*$ if for every $x, y \in \mathcal{K}$ we have $\left| f(y) - f(x) - \nabla f(x)^\top (y - x) \right| \leq \frac{L}{2} \| x - y \|_*^2$. We define the projection $\Gamma : \mathbb{R}^{|A|} \to \Delta_A$, with respect to a set $A$, where $\Delta_A$ denotes the probability simplex over $A$. For any $x \in \mathbb{R}^{|A|}$, $\Gamma[x] = \arg\min_{y \in \Delta_A} \| x - y \|$. An important generalization of the property of convexity we use henceforth is that of gradient domination.

**Definition 1** (Gradient Domination). A function $f : \mathcal{K} \to \mathbb{R}$ is said to be $(\kappa, \tau, \mathcal{K}_1, \mathcal{K}_2)$-locally gradient dominated (around $\mathcal{K}_1$ by $\mathcal{K}_2$) if for all $x \in \mathcal{K}_1$, it holds that

$$\max_{y \in \mathcal{K}} f(y) - f(x) \ \leq \ \kappa \cdot \max_{y \in \mathcal{K}_2} \left\{ \nabla f(x)^\top (y - x) \right\} + \tau.$$

**Markov decision process.**  An infinite-horizon discounted Markov Decision Process (MDP) $\mathcal{M} = (S, A, P, r, \gamma, d_0)$ is specified by: a state space $S$, an action space $A$, a transition model $P$ where $P(s'|s, a)$ denotes the probability of immediately transitioning to state $s'$ upon taking action $a$ at state $s$, a reward function $r : S \times A \to [0, 1]$ where $r(s, a)$ is the immediate reward associated with taking action $a$ at state $s$, a discount factor $\gamma \in [0, 1)$; a starting state distribution $d_0$ over $S$. For any infinite-length state-action sequence (hereafter, called a trajectory), we assign the following value $V(\varsigma = (s_0, a_0, s_1, a_1, \dots)) = \sum_{t=0}^{\infty} \gamma^t r(s_t, a_t)$. The agent interacts with the MDP through the choice of stochastic policy $\pi : S \to \Delta_A$ it executes. The execution of such a policy induces a distribution over trajectories $\varsigma = (s_0, a_0, \dots)$ as $P(\varsigma|\pi) = d_0(s_0) \prod_{t=0}^{\infty} (P(s_{t+1}|s_t, a_t) \pi(a_t|s_t))$. Using this description we can associate a state $V^\pi(s)$ and state-action $Q^\pi(s, a)$ value function with any policy $\pi$. For an arbitrary distribution $d$ over $S$, define:

$$Q^\pi(s, a) = \mathbb{E} \left[ \sum_{t=0}^{\infty} \gamma^t r(s_t, a_t) \middle| \pi, s_0 = s, a_0 = a \right],$$

$$V^\pi(s) = \mathbb{E}_{a \sim \pi(\cdot|s)} \left[ Q^\pi(s, a) | \pi, s \right], \ V_d^\pi = \mathbb{E}_{s_0 \sim d} \left[ V^\pi(s) | \pi \right].$$

Here the expectation is with respect to the randomness of the trajectory induced by $\pi$ in $\mathcal{M}$. When convenient, we shall use $V^\pi$ to denote $V_{d_0}^\pi$, and $V^*$ to denote $\max_\pi V^\pi$.

Similarly, to any policy $\pi$, one may ascribe a (discounted) state-visitation distribution $d^\pi = d_{d_0}^\pi$.

$$d_d^\pi(s) = (1 - \gamma) \sum_{t=0}^{\infty} \gamma^t \sum_{\varsigma : s_t = s} P(\varsigma|\pi, s_0 \sim d)$$

**Modes of Accessing the MDP.**  We henceforth consider two modes of accessing the MDP, that are standard in the reinforcement learning literature, and provide different results for each.

The first natural access model is called the **episodic rollout setting.** This mode of interaction allows us to execute a policy, stop and restart at any point, and do this multiple times.

Another interaction model we consider is called **rollout with $\nu$-restarts.** This is similar to the episodic setting, but here the agent may draw from the MDP a trajectory seeded with an initial state distribution $\nu \neq d_0$. This interaction model was considered in prior work on policy optimization [24, 2]. The motivation for this model is two-fold: first, $\nu$ can be used to incorporate priors (or domain knowledge) about the state coverage of the optimal policy; second, $\nu$ provides a mechanism to incorporate exploration into policy optimization procedures.

### 2.1  Weak learning

Our boosting algorithms henceforth call upon weak learners to generate weak policies. We formalize the notion of a weak learner next. We consider two types of weak learners, and give different end results based on the different assumptions: weak supervised and weak online learners. In the discussion below, let $\pi_{Rand}$ be a uniformly random policy, i.e. $\forall (s, a) \in S \times A, \pi_{Rand}(a|s) = 1/|A|$. The formal definition and results for the online setting are deferred to the appendix. In what follows we define the supervised weak learning model.

The natural way to define weak learning is an algorithm whose performance is always slight better than that of random policy, one that chooses an action uniformly at random at any given state. However, in general no learner can outperform a random learner over all label distributions. This motivates the literature on agnostic boosting [25, 9, 19] that defines a weak learner as one that can approximate the best policy in a given policy class.

**Definition 2** (Weak Supervised Learner). Let $\alpha \in (0, 1]$. Consider a class $\mathcal{L}$ of linear loss functions $\ell : \mathbb{R}^A \to \mathbb{R}$, a family $\mathbb{D}$ of distributions that are supported over $S \times \mathcal{L}$, and policy class $\Pi$. A weak supervised learning algorithm, for every $\varepsilon, \delta > 0$, given $m(\varepsilon, \delta) = \frac{\log |\mathcal{W}|}{\varepsilon^2} \log \frac{1}{\delta}$ samples $D_m$ from any distribution $\mathcal{D} \in \mathbb{D}$ outputs a policy $\mathcal{W}(D_m) \in \Pi$ such that with probability $1 - \delta$,

$$\mathbb{E}_{(s,\ell) \sim \mathcal{D}}\big[\ell(\mathcal{W}(D_m))\big] \ \leq \ \alpha \min_{\pi^* \in \Pi} \mathbb{E}_{(s,\ell) \sim \mathcal{D}}\big[\ell(\pi^*(s))\big] + (1 - \alpha) \mathbb{E}_{(s,\ell) \sim \mathcal{D}}\big[\ell(\pi_{Rand}(s))\big] + \varepsilon.$$

Note that the weak learner outputs a policy in $\Pi$ which is approximately competitive against the class $\Pi$. As an additional relaxation, instead of requiring that the weak learning guarantee holds for all distributions, in our setup, it will be sufficient that the weak learning assumption holds over *natural* distributions. Specifically, we define a class of *natural* distributions $\mathbb{D}$, such that $\mathcal{D} \in \mathbb{D}$ if and only if there exists some $\pi \in \Pi$ such that, $\mathcal{D}(s) = \int_\ell \mathcal{D}(s, \ell) d\mu(\ell) = d^\pi(s)$. In particular, while a *natural* distribution may have arbitrary distribution over labels, its marginal distribution over states must be realizable as the state distribution of some policy in $\Pi$ over the MDP $\mathcal{M}$. Therefore, the complexity of weak learning adapts to the complexity of the MDP itself. As an extreme example, in stochastic contextual bandits where policies do not affect the distribution of states (say $d_0$), it is sufficient that the weak learning condition holds with respect to all couplings of a single distribution $d_0$.

## 3 Algorithm & Main Results

In this section we describe our RL boosting algorithm. Here we focus on the case where a supervised weak learning is provided. The online weak learners variant of our result is detailed in the appendix. We next define several definitions and algorithmic subroutines required for our method.

### 3.1 Policy aggregation

For a base class of policies $\Pi$, our algorithm incrementally builds a more expressive policy class by aggregating base policies via both linear combinations and non-linear transformations. In effect, the algorithm produces a finite-width depth-2 circuit over some subset of the base policy class. That is, our approach can be thought of as an aggregation of base policies, which forms a 2-layer neural network, as depicted in Figure 1. The leaves of the tree are the policies $\pi \in \Pi$ the base policy class. These are then linearly aggregated to form the first layer of the tree, denoted $\tilde{\pi}_1, \tilde{\pi}_2$ in Figure 1.

Next, each linear combination of policies in the overall aggregation undergoes a projection operation. The projection may be viewed as a non-linear activation function, such as ReLU, in deep learning terms. Note that the projection of any function from $S$ to $\mathbb{R}^{|A|}$ produces a policy, i.e. a mapping from states to distributions over actions. In the analysis of our algorithm we give a particular projection operation $\Gamma[\cdot]$ which allows us to yield the desired guarantees.

**Definition 3** (Policy Projection). Given $\tilde{\pi} : S \to \mathbb{R}^{|A|}$, define a projected policy $\pi = \Gamma[\tilde{\pi}]$ to be a policy such that simultaneously for all $s \in S$, it holds that $\pi(\cdot|s) = \Gamma\left[\tilde{\pi}(s)\right]$.

**Definition 4** (Policy Tree). A *Policy Tree* $\Pi \subseteq S \to \Delta_A$ with respect to $\Pi \subseteq S \to \Delta_A$ some base policy class, and $N, T \in \mathbb{N}$, is a linear combination of $T$ projected policies $\Gamma[\tilde{\pi}]$, where each $\tilde{\pi}$ is a linear combination of $N$ base policies $\pi \in \Pi$.

This final definition describes the set of possible outputs of the boosting procedure. It is important that the policy that the boosting algorithm outputs can be evaluated efficiently. In the appendix we show it is indeed the case (see Lemma 12). Hereafter, we refer to a Policy Tree with respect to $\Pi$, $N$ and $T$, as $\Pi$ for $N, T = O(\text{poly}(|A|, (1 - \gamma)^{-1}, \varepsilon^{-1}, \alpha^{-1}, \log \delta^{-1}))$ specified later.

### 3.2 Main results

Next, we give the main results of our RL boosting algorithm via weak supervised learning, specified in Algorithm 1.

**A Policy Tree $\bar{\pi}$**

Figure 1: The figure illustrates a Policy Tree hierarchy (see Definition 4), output of the boosting procedure specified in Algorithm 1. Specifically, it is obtained by setting $N = 3$ on the inner loop of Internal Boost (Algorithm 2), and $T = 2$ on the main booster (Algorithm 1). Overall we get all base policies $\pi_1, ..., \pi_6 \in \Pi$ on the lower level, to form the Policy Tree $\bar{\pi} \in \mathbb{\Pi}$.

---

**Algorithm 1** RL Boosting

---

1: **Input**: number of iterations $T$, initial state distribution $\mu$, and $P, N, M$ parameters for Internal Boost.
2: Initialize a policy $\pi_0 \in \Pi$ arbitrarily.
3: **for** $t = 1$ **to** $T$ **do**
4:    Run Internal Boost (Algorithm 2) with distribution $\mu$ and policy $\pi_t$ to obtain $\pi_t'$.
5:    Update $\pi_t = (1 - \eta_{1,t})\pi_{t-1} + \eta_{1,t}\pi_t'$.
6: **end for**
7: Run each policy $\pi_t$ for $P$ rollouts to compute an empirical estimate $\widehat{V^{\pi_t}}$ of the expected return.
8: **return** $\bar{\pi} := \pi_{t'}$ where $t' = \arg\max_t \widehat{V^{\pi_t}}$.

---

To state the results, we need the following definitions. The first generalizes the policy completeness notion from [33]. It may be seen as the policy-equivalent analogue of inherent bellman error [29]. Intuitively, it measures the degree to which a policy in $\Pi$ can best approximate the bellman operator in an average sense with respect to the state distribution induced by a policy from $\mathbb{\Pi}$.

**Definition 5** (Policy Completeness). For any initial state distribution $\mu$, and policy classes $\Pi, \mathbb{\Pi}$, define $\mathcal{E}_\mu = \max_{\pi \in \mathbb{\Pi}} \min_{\pi^* \in \Pi} \mathbb{E}_{s \sim d_\mu^\pi} \left[ \max_{a \in A} Q^\pi(s, a) - Q^\pi(s, \cdot)^\top \pi^*(\cdot | s) \right]$.

**Definition 6** (Distribution Mismatch). Let $\pi^* = \arg\max_\pi V^\pi$, and $\nu$ a fixed initial state distribution (see section 2). Define the following distribution mismatch coefficients: $C_\infty = \max_{\pi \in \mathbb{\Pi}} \left\| d^{\pi^*} / d^\pi \right\|_\infty$, $D_\infty = \left\| d^{\pi^*} / \nu \right\|_\infty$.

The above notion of the distribution mismatch coefficient is often useful to characterize the exploration problem faced by policy optimization algorithms. We now give the main result for the output of our RL boosting algorithm, assuming supervised weak learners.

**Theorem 7.** *Algorithm 1 samples $T(MN + P)$ episodes of length $\tilde{O}(\frac{1}{1-\gamma})$ with probability $1 - \delta$.*
*In the __episodic model__, with $\mu = d_0$, for $\eta_{1,t} = \min\{1, \frac{2C_\infty}{t}\}$, $T = O\left(\frac{C_\infty^2}{(1-\gamma)^3 \varepsilon}\right)$, $N = \left(\frac{16|A|C_\infty}{(1-\gamma)^2 \alpha \epsilon}\right)^2$,*
*$M = m\left(\frac{(1-\gamma)^2 \alpha \varepsilon}{C_\infty |A|}, \frac{\delta}{NT}\right)$, with probability $1 - \delta$, $V^* - V^\pi \leq \frac{C_\infty \mathcal{E}}{1-\gamma} + \varepsilon$.*
*In the __$\nu$-reset model__, with $\mu = \nu$, for $\eta_{1,t} = \sqrt{\frac{8\gamma(1-\gamma)^2}{|A|^2 T}}$, $T = \frac{8D_\infty^2}{(1-\gamma)^6 \varepsilon^2}$, $N = \left(\frac{16|A|D_\infty}{(1-\gamma)^3 \alpha \epsilon}\right)^2$,*
*$M = m\left(\frac{(1-\gamma)^3 \alpha \varepsilon}{8|A|D_\infty}, \frac{\delta}{2NT}\right)$, with probability $1 - \delta$, $V^* - V^\pi \leq \frac{D_\infty \mathcal{E}_\nu}{(1-\gamma)^2} + \varepsilon$.*
***Sample complexities:*** *If $m(\varepsilon, \delta) = \frac{\log|\mathcal{W}|}{\varepsilon^2} \log \frac{1}{\delta}$ for some measure of weak learning complexity $|\mathcal{W}|$,*

---

**Algorithm 2** Internal Boost

---
1: **Input**: number of iterations $N$, number of episodes $M$, initial policy $\pi$, initial state distribution $\mu$.
2: Set $\tilde{\pi}_0$ to be an arbitrary policy in $\Pi$.
3: **for** $n = 1$ **to** $N$ **do**
4:     Execute $\pi$ with $\mu$ via Algorithm 3 for $M$ episodes, to get $D_n = \{(s_i, \widehat{Q}_i)_{i=1}^M\}$.
5:     Modify $D_n$ to produce a new dataset $D'_n = \{(s_i, f_i)\}_{i=1}^M$, such that for all $i \in [m]$:

$$f_i = \frac{1}{\beta}\left(y_i - \tilde{\pi}_n(\cdot|s_i)\right), \quad y_i = \arg\min_{y \in \mathbb{R}^{|A|}}\{-\widehat{Q}_i^\top y + G \min_{z \in \Delta_A}\|z - y\| + \frac{\|\tilde{\pi}_n(\cdot|s_i) - y\|^2}{2\beta}\}$$

    where $G = \frac{A}{1-\gamma}, \beta = \frac{2\gamma}{(1-\gamma)^3}$ and $f_i, \widehat{Q}_i \in \mathbb{R}^{|A|}$.
6:     Let $\mathcal{A}_n$ be the policy chosen by the weak learning oracle when given data set $D'_{t,n}$.
7:     Update

$$\tilde{\pi}_n = (1 - \eta_{2,n})\tilde{\pi}_{n-1} + \frac{\eta_{2,n}}{\alpha}\mathcal{A}_n.$$

8: **end for**
9: **return** $\Gamma[\tilde{\pi}_N]$.

---

---

**Algorithm 3** Trajectory Sampler: samples a state $s \sim d^\pi$, and an unbiased estimate of $Q_s^\pi$

---
1: Sample state $s_0 \sim \mu$, action $a' \sim \mathcal{U}(A)$ uniformly.
2: Sample $s \sim d^\pi$ as follows: at every timestep $h$, with probability $\gamma$, act according to $\pi$; else, accept $s_h$ as the sample and proceed to Step 3.
3: Take action $a'$ at state $s_h$, then continue to execute $\pi$, and use a termination probability of $1 - \gamma$. Upon termination, set $R(s_h, a')$ as the *undiscounted* sum of rewards from time $h$ onwards.
4: Define the vector $\widehat{Q_{s_h}^\pi}$, such that for all $a \in A$, $\widehat{Q_{s_h}^\pi}(a) = |A| \cdot R(s_h, a') \cdot \mathbb{I}_{a=a'}$.
5: **return** $(s_h, \widehat{Q_{s_h}^\pi})$.

---

*the algorithm samples $\tilde{O}\left(\frac{C_\infty^6 |A|^4 \log|\mathcal{W}|}{(1-\gamma)^{11}\alpha^4\varepsilon^5}\right)$ episodes in the episodic model, and $\tilde{O}\left(\frac{D_\infty^6 |A|^4 \log|\mathcal{W}|}{(1-\gamma)^{18}\alpha^4\varepsilon^6}\right)$ in the $\nu$-reset model.*

Theorem 7 above pertains to the case where a weak learning algorithm is available. However, another main result is given by considering the simpler approach of reduction of RL to a *strong* supervised learning algorithm. In particular, when running our main boosting algorithm, we can replace the call to Internal Boost (in Line 4 of Algorithm 1) with a call to a *strong* supervised learning algorithm. By a similar analysis to that of Theorem 7 we obtain the following corollary.

**Corollary 8.** *Let $m(\varepsilon, \delta) = \frac{\log|\mathcal{W}|}{\varepsilon^2}\log\frac{1}{\delta}$ for some measure of weak learning complexity $|\mathcal{W}|$. When run with a supervised learning oracle (Definition 2) with $\alpha = 1$, i.e. $N = 1$) as the Internal boosting, Algorithm 1 samples $\tilde{O}\left(\frac{C_\infty^3 \log|\mathcal{W}|}{\varepsilon^3}\right)$ episodes in the episodic model, and $\tilde{O}\left(\frac{D_\infty^4 \log|\mathcal{W}|}{\varepsilon^4}\right)$ in the $\nu$-reset model, to guarantee $V^* - V^\pi \leq \frac{C_\infty\varepsilon}{1-\gamma} + \varepsilon$ with probability $1 - \delta$ in the episodic model and $V^* - V^\pi \leq \frac{D_\infty\varepsilon_\nu}{(1-\gamma)^2} + \varepsilon$ in the $\nu$-reset model.*

We note that this result is an improvement over previous results in terms of sample complexity requirement of the algorithm. In particular, in [24], Theorem 4.4 and Corollary 4.5 achieve the same guarantee using $O(1/\varepsilon^4)$ samples regardless of the MDP access model. Briefly, CPI utilizes $1/\varepsilon^2$ calls to an $\varepsilon$-optimal supervised learning oracle (each call needing $1/\varepsilon^2$ samples) to reach a $\varepsilon$-local optima of the value function. Under requisite state coverage assumptions, this translates to $\varepsilon$-function value suboptimality. Indeed, such mode of analysis via first arguing for convergence to a local optima for the CPI algorithm can be shown to be tight. The improvement in our case for the episodic access model comes from the insight that it is possible to make direct claims on the function value sub-optimality (second part of Theorem 9), bypassing the need for making a claim on the local optimality, in the gradient-dominated case.

### 3.3 Trajectory sampler

In Algorithm 3 we describe an episodic sampling procedure, that is used in our sample-based RL boosting algorithms described above. For a fixed initial state distribution $\mu$, and any given policy $\pi$, we apply the following sampling procedure: start at an initial state $s_0 \sim \mu$, and continue to act thereafter in the MDP according to any policy $\pi$, until termination. With this process, it is straightforward to both sample from the state visitation distribution $s \sim d^\pi$, and to obtain unbiased samples of $Q^\pi(s, \cdot)$; see Algorithm 3 for the detailed process.

## 4 Sketch of the analysis

**Non-convex Frank-Wolfe.** We give an abstract high-level procedural template that the previously introduced RL boosters operate in. This is based on a variant of the Frank-Wolfe optimization technique [15], adapted to non-convex and gradient dominated function classes (see Definition 1). The Frank-Wolfe (FW) method assumes oracle access to a black-box linear optimizer, denoted $\mathcal{O}$, and utilizes it by iteratively making oracle calls with modified objectives, in order to solve the harder task of convex optimization. Analogously, boosting algorithms often assume oracle access to a "weak" learner, which are utilized by iteratively making oracle calls with modified objective, in order to obtain a "strong" learner, with boosted performance. In the RL setting, the objective is in fact non-convex, but exhibits gradient domination. By adapting Frank-Wolfe technique to this setting, we will in subsequent section obtain guarantees for the algorithms given in Section 3. **Oracle:** Denote by $\mathcal{O}$ a black-box oracle to an $(\epsilon_0, \mathcal{K}_2)$-approximate linear optimizer over a convex set $\mathcal{K} \subseteq \mathbb{R}^d$ such that for any given $v \in \mathbb{R}^d$, we have $v^\top \mathcal{O}(v) \geq \max_{u \in \mathcal{K}_2} v^\top u - \epsilon_0$.

---

**Algorithm 4** Non-convex Frank-Wolfe

1: Input: $T > 0$, objective $f$, linear optimizer $\mathcal{O}$, rate $\eta_t$.
2: Choose $x_0 \in \mathcal{K}$ arbitrarily.
3: **for** $t = 1, \ldots, T$ **do**
4:     Call $z_t = \mathcal{O}(\nabla_{t-1})$, where $\nabla_{t-1} = \nabla f(x_{t-1})$. Set $x_t = (1 - \eta_t)x_{t-1} + \eta_t z_t$.
5: **end for**
6: **return** $\bar{x} := x_{t'}$ where $t' = \arg\min_t \nabla_t^\top (z_t - x_t)$.

---

**Theorem 9.** *Let $f : \mathcal{K} \to \mathbb{R}$ be $L$-smooth in some norm $\|\cdot\|_*$, bounded for all $x \in \mathcal{K}$, $|f(x)| \leq H$ for some $H > 0$, and let the diameter of $\mathcal{K}$ in $\|\cdot\|_*$ be $D$. Then, for a $(\epsilon_0, \mathcal{K}_2)$-linear optimization oracle $\mathcal{O}$, and $\eta_t = \eta = \sqrt{\frac{4H}{LD^2 T}}$, the output $\bar{x}$ of Algorithm 4 satisfies*

$$\max_{u \in \mathcal{K}_2} \nabla f(\bar{x})^\top (u - \bar{x}) \leq \sqrt{\frac{2HLD^2}{T}} + \epsilon_0; \quad \max_{x^* \in \mathcal{K}} f(x^*) - f(\bar{x}) \leq \frac{2\kappa^2 \max\{LD^2, H\}}{T} + \tau + \kappa\epsilon_0$$

*Furthermore, if $f$ is $(\kappa, \tau, \mathcal{K}_1, \mathcal{K}_2)$-locally gradient-dominated and $x_0, \ldots x_T \in \mathcal{K}_1$, then the output $\bar{x}$ of Algorithm 4 where $\eta_t = \min\{1, \frac{2\kappa}{t}\}$ satisfies the bound on the right.*

We sketch the high-level ideas of the proof of our main result, stated in Theorem 7, and refer the reader to the appendix for the formal proof. We will establish an equivalence between RL Boosting (Algorithm 1) and the variant of the Frank-Wolfe algorithm (Algorithm 4). This abstraction allows us to obtain the novel convergence guarantees given in Theorem 7. Throughout the analysis, we use the notation $\nabla_\pi V^\pi$ to denote the gradient of the value function with respect to the $|S| \times |A|$-sized representation of the policy $\pi$, namely the functional gradient of $V^\pi$.

**Internal-boosting weak learners.** The Frank-Wolfe algorithm utilizes an inner gradient optimization oracle as a subroutine. To implement this oracle using approximate optimizers, we utilize yet another variant of the FW method as "internal-boosting" for the weak learners, by employing an adapted analysis of [19] that is stated in Claim 10 below. Let $\mathcal{D}_t$ be the distribution induced by the trajectory sampler in round $t$.

**Claim 10.** *Let $\beta = \sqrt{1/\alpha N}$, $\eta_{2,n} = \min\{2/n, 1\}$. $\pi'_t$ produced by Algorithm 1 satisfies*
$$\max_{\pi \in \Pi} \mathbb{E}_{(s,Q) \sim \mathcal{D}_t} \left[ Q^\top \pi(s) \right] - \mathbb{E}_{(s,Q) \sim \mathcal{D}_t} \left[ Q^\top \pi'_t(s) \right] \leq (2|A|/(1-\gamma)\alpha) \left( \varepsilon + 2/\sqrt{N} \right).$$

**From weak learning to linear optimization,** Next, we give an important observation which allows us to re-state the guarantee in the previous subsection in terms of linear optimization over functional gradients. The key observation here is that the expensive optimizing procedure for $(\nabla_\pi V^\pi)^\top \pi'$, which in particular requires iterating over all states in $S$, can be instead replaced with sampling from an appropriate distribution $\mathcal{D}$ (via Algorithm 3). These sample pairs $(s, \widehat{Q^\pi}(s, \cdot))$ could then be fed to our weak learning algorithm, which guarantees generalization.

**Lemma 11.** *Applying Algorithm 3 for any given policy $\pi$ yields an unbiased estimate of the gradient, such that for any $\pi'$, $(\nabla_\pi V_\mu^\pi)^\top \pi' = \mathbb{E}_{(s, \widehat{Q^\pi}(s, \cdot)) \sim \mathcal{D}} \left[ \widehat{Q^\pi}(s, \cdot)^\top \pi'(\cdot|s) \right]/(1 - \gamma)$, where $\pi'(\cdot|s) \in \Delta_A$, $\mathcal{D}$ is the distribution induced on the outputs of Algorithm 3, for the policy $\pi$ and initial distribution $\mu$.*

## 5   Experiments

The primary contribution of the present work is theoretical. Nevertheless, we empirically test our proposal with the experiment designed to elicit qualitative properties of the proposed algorithm, instead of aiming to achieve the state-of-the-art. To validate our results, we check if the proposed algorithm is indeed capable of boosting the accuracy of concrete instantiations of weak learners. We use depth-3 decision trees, with the implementation adapted from Scikit-Learn [30], as our base weak learner. This choice of weak learner is particularly suitable for boosting, because it is an impoverished policy class in a representational sense and hence it is reasonable to expect that it may do only slightly better than random guessing with respect to the classification loss. We consider the performance of the boosting algorithm (Algorithm 1) across multiple rounds of boosting or number of weak learners to that of supervised-learning-based policy iteration; the computational burden of the algorithm scales linearly with the latter. Throughout all the experiments, we used $\eta = 0.9$. To speed up computation, the plots below were generated by retaining the 3 most recent policies of every iteration in the policy mixture. We evaluated these on the CartPole and the LunarLander environments. The results demonstrate the proposed RL boosting algorithm succeeds in maximizing rewards while using few weak learners (equivalently, within a few rounds of boosting).

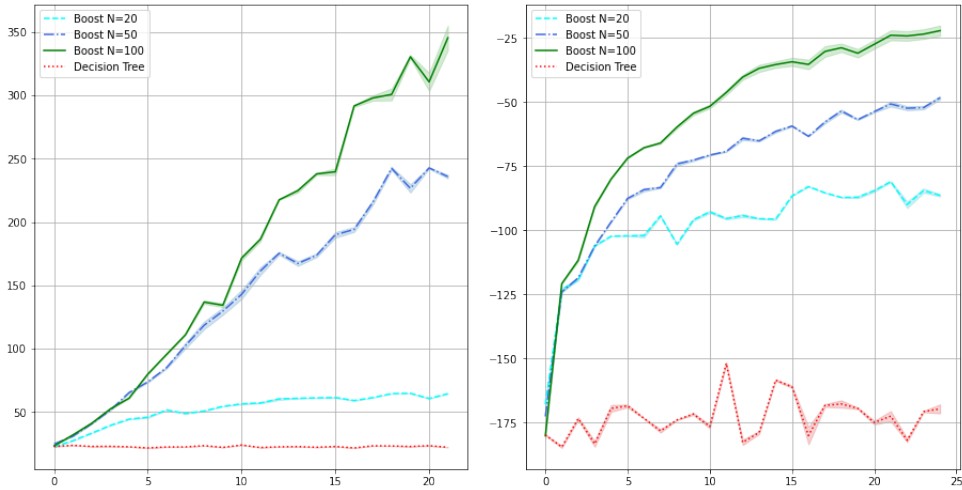

Figure 2: Reward trajectory for the CartPole (left) and the LunarLander (right) environments of the proposed boosting algorithm for $N = 20, 50, 100$ number of base weak learners is compared to supervised-learning-based policy iteration (decision tree) above. The x-axis corresponds to $T$ number of iterations, and for each $t \in [T]$, reward is computed over 100 episodes of interactions. The confidence interval is plotted over 3 such runs.

## 6   Conclusions

Building on recent advances in boosting for online convex optimization and bandits, we have described a boosting algorithm for reinforcement learning over large state spaces with provable guarantees. We see this as a first attempt at using a tried-and-tested methodology from supervised learning to RL.

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
