# A  Notation: List of Symbols

**Weak Learning and Boosting**

| | |
|---|---|
| $\alpha$ | Weak learning parameter |
| $T$ | Number of boosting iterations |
| $N$ | Number of internal-boosting iterations |
| $M$ | Number of internal-boosting episodes |
| $\Gamma[\cdot]$ | Policy projection |
| $\Pi$ | Policy class |
| $\mathbb{\Pi}$ | Policy-Tree class (w.r.t $\Pi$, $\Gamma$, $N$ and $T$) |

**Markov Decision Process**

| | |
|---|---|
| $S$ | State space |
| $A$ | Action space |
| $\Delta_A$ | Probability simplex over actions |
| $Q^\pi(s, a)$ | Q function |
| $V^\pi(s)$ | Value function |
| $d(s_0)$ | Initial state distribution |
| $d_d^\pi(s)$ | State-visitation distribution w.r.t $\pi, d$ |
| $\gamma$ | Discount factor |
| $\mathcal{E}_\mu(\mathbb{\Pi}, \Pi)$ | Policy completeness |
| $\mu, \nu$ | Used for different initial state distributions |
| $C_\infty$ | Distribution mismatch if $\mu = d_0$ |
| $D_\infty$ | Distribution mismatch if $\mu = \nu \neq d_0$ |

**Optimization**

| | |
|---|---|
| $\mathcal{K}$ | Decision set |
| $L$ | Smoothness of the objective |
| $H$ | Upper bound on the range of function value |
| $D$ | Upper bound on Euclidean diameter |

# B  Appendix

It is important that the policy that the boosting algorithm outputs can be evaluated efficiently. Towards that end, we give the following claim.

**Claim 12.** *For any $\pi \in \mathbb{\Pi}(\Pi, N, T)$, $\pi(\cdot|s)$ for any $s \in S$ can be evaluated using $TN$ base policy evaluations and $O(T \times (NA + A \log A))$ arithmetic and logical operations.*

*Proof.* Since $\pi \in \mathbb{\Pi}(\Pi, N, T)$, it is composed of $TN$ base policies. Producing each aggregated function takes $NA$ additions and multiplications; there are $T$ of these. Each projection takes time equivalent to sorting $|A|$ numbers, due to a water-filling algorithm [14]; these are also $T$ in number. The final linear transformation takes an additional $TA$ operations. $\square$

# C    RL Boosting via Weak Online Learning

The second model of weak learning we consider requires a stronger assumption, but will give us better sample and oracle complexity bounds henceforth.

**Definition 13** (Weak Online Learner). Let $\alpha \in (0,1)$. Consider a class $\mathcal{L}$ of linear loss functions $\ell : \mathbb{R}^A \to \mathbb{R}$. A weak online learning algorithm, for every $M > 0$, incrementally for each timestep computes a policy $\mathcal{W}_m \in \Pi$ and then observes the state-loss pair $(s, \ell_t) \in S \times \mathcal{L}$ such that

$$\sum_{m=1}^{M} \ell_m(\mathcal{W}_m(s_m)) \geq \alpha \max_{\pi^* \in \Pi} \sum_{m=1}^{M} \ell_m(\pi^*(s_m)) + (1-\alpha) \sum_{m=1}^{M} \ell_m(\pi_{Rand}(s_m)) - R_{\mathcal{W}}(M).$$

**Assumption 1** (Weak Online Learning). *The booster has access to a weak online learning oracle (Definition 13) over the policy class $\Pi$, for some $\alpha \in (0,1)$.*

**Remark 14.** A similar remark about *natural* distributions applies to the online weak learner. In particular, it is sufficient the guarantee in 13 holds for arbitrary sequence of loss functions with high probability over the sampling of the state from $d^\pi$ for some $\pi \in \Pi$. Although stronger than supervised weak learning, this oracle can be interpreted as a relaxation of the online weak learning oracle considered in [9, 10, 19]. A similar model of hybrid adversarial-stochastic online learning was considered in [31, 26, 8]. In particular, it is known [26] that unlike online learning, the capacity of a hypothesis class for this model is governed by its VC dimension (vs. Littlestone dimension).

---

**Algorithm 5** RL Boosting via Weak Online Learning

---
1: Initialize a policy $\pi_0 \in \Pi$ arbitrarily.
2: **for** $t = 1$ **to** $T$ **do**
3:     Initialize online weak learners $\mathcal{W}_1, \ldots \mathcal{W}^N$.
4:     **for** $m = 1$ **to** $M$ **do**
5:         Execute $\pi_{t-1}$ once with initial state distribution $\mu$ via Algorithm 3, to get $(s_{t,m}, \widehat{Q}_{t,m})$.
6:         Choose $\tilde{\pi}_{t,m,0} \in \Pi$ arbitrarily.
7:         **for** $n = 1$ **to** $N$ **do**
8:             Set $\tilde{\pi}_{t,m,n} = (1 - \eta_{2,n})\tilde{\pi}_{t,m,n-1} + \frac{\eta_{2,n}}{\alpha}\mathcal{W}^n$.
9:         **end for**
10:        Pass to each $\mathcal{W}^n$ the following loss linear $f_{t,m,n}$:

$$f_{t,m,n} = \frac{1}{\beta}\left(y_{t,m,n} - \tilde{\pi}_{t,m,n}(\cdot|s_i)\right).$$

where $G = \frac{A}{1-\gamma}, \beta = \frac{2\gamma}{(1-\gamma)^3}$ and $f_i, \widehat{Q}_i \in \mathbb{R}^{|A|}$

$$y_i = \arg\min_{y \in \Delta_A}\{-\widehat{Q}_{t,m}^\top y + G \min_{z \in \Delta_A} \|z - y\| + \frac{\|\tilde{\pi}_{t,m,n}(\cdot|s_{t,m}) - y\|^2}{2\beta}\}$$

11:    **end for**
12:    Declare $\pi_t' = \frac{1}{M}\sum_{m=1}^{M} \Gamma\left[\tilde{\pi}_{t,m,N}\right]$.
13:    Choose $\eta_{1,t} = \min\{1, \frac{2C_\infty}{t}\}$ if $\mu = d_0$ else set $\eta_{1,t} = \sqrt{\frac{8\gamma(1-\gamma)^2}{|A|^2 T}}$.
14:    Update $\pi_t = (1 - \eta_{1,t})\pi_{t-1} + \eta_{1,t}\pi_t'$.
15: **end for**
16: Run each policy $\pi_t$ for $P$ rollouts to compute an empirical estimate $\widehat{V^{\pi_t}}$ of the expected return.
17: **return** $\bar{\pi} := \pi_{t'}$ where $t' = \arg\max_t \widehat{V^{\pi_t}}$.

---

**Theorem 15.** *Algorithm 5 samples $TM$ episodes of length $\frac{1}{1-\gamma}\log\frac{TM}{\delta}$ with probability $1 - \delta$. In the episodic model, Algorithm 5 guarantees as long as $T = \frac{16C_\infty^2}{(1-\gamma)^3\varepsilon}$, $N = \left(\frac{16|A|C_\infty}{(1-\gamma)^2\alpha\epsilon}\right)^2$, $M = \max\left\{\frac{1000|A|^2C_\infty^2}{(1-\gamma)^4\varepsilon^2\alpha^2}\log^2 T\delta, \frac{8|A|C_\infty R_{\mathcal{W}}(M)}{(1-\gamma)^2\alpha\varepsilon}\right\}, \mu = d_0$, we have with probability $1 - \delta$*

$$V^* - V^\pi \leq C_\infty \frac{\mathcal{E}(\Pi, \Pi)}{1 - \gamma} + \varepsilon$$

In the $\nu$-reset model, Algorithm 1 guarantees as long as $T = \frac{100D_\infty^2}{(1-\gamma)^6\varepsilon^2}$, $N = \left(\frac{20|A|D_\infty}{(1-\gamma)^3\alpha\epsilon}\right)^2$, $M = \max\left\{\left(\frac{40|A|D_\infty}{(1-\gamma)^3\alpha\varepsilon}\log\frac{T}{\delta}\right)^2, \frac{10|A|D_\infty R_\mathcal{W}(M)}{(1-\gamma)^3\alpha\varepsilon}\right\}$, $\mu = \nu$, we have with probability $1 - \delta$

$$V^* - V^\pi \;\leq\; D_\infty\frac{\mathcal{E}_\nu(\sqcap, \Pi)}{(1 - \gamma)^2} + \varepsilon$$

If $R_\mathcal{W}(M) = \sqrt{M\log|\mathcal{W}|}$ for some measure of weak learning complexity $|\mathcal{W}|$, the algorithm samples $\tilde{O}\left(\frac{C_\infty^4|A|^2\log|\mathcal{W}|}{(1-\gamma)^7\alpha^2\varepsilon^3}\right)$ episodes in the episodic model, and $\tilde{O}\left(\frac{D_\infty^4|A|^2\log|\mathcal{W}|}{(1-\gamma)^{12}\alpha^2\varepsilon^4}\right)$ in the $\nu$-reset model.

# D Analysis for Boosting with Weak Supervised Learning (Proof of Theorem 7)

**Theorem** (Formal version of Theorem 7). *Algorithm 1 samples $TMN$ episodes of length $\frac{1}{1-\gamma}\log\frac{TMN}{\delta}$ with probability $1 - \delta$. In the episodic model, Algorithm 1 guarantees as long as $T = \frac{16C_\infty^2)}{(1-\gamma)^3\varepsilon}$, $N = \left(\frac{16|A|C_\infty}{(1-\gamma)^2\alpha\epsilon}\right)^2$, $M = m\left(\frac{(1-\gamma)^2\alpha\varepsilon}{8C_\infty|A|}, \frac{\delta}{NT}\right)$, $\mu = d_0$, we have with probability $1 - \delta$*

$$V^* - V^\pi \;\leq\; C_\infty\frac{\mathcal{E}(\sqcap, \Pi)}{1 - \gamma} + \varepsilon$$

*In the $\nu$-reset model, Algorithm 1 guarantees as long as $T = \frac{8D_\infty^2}{(1-\gamma)^6\varepsilon^2}$, $N = \left(\frac{16|A|D_\infty}{(1-\gamma)^3\alpha\epsilon}\right)^2$, $M = m\left(\frac{(1-\gamma)^3\alpha\varepsilon}{8|A|D_\infty}, \frac{\delta}{2NT}\right)$, $\mu = \nu$, we have with probability $1 - \delta$*

$$V^* - V^\pi \;\leq\; D_\infty\frac{\mathcal{E}_\nu(\sqcap, \Pi)}{(1 - \gamma)^2} + \varepsilon$$

*If $m(\varepsilon, \delta) = \frac{\log|\mathcal{W}|}{\varepsilon^2}\log\frac{1}{\delta}$ for some measure of weak learning complexity $|\mathcal{W}|$, the algorithm samples $\tilde{O}\left(\frac{C_\infty^6|A|^4\log|\mathcal{W}|}{(1-\gamma)^{11}\alpha^4\varepsilon^5}\right)$ episodes in the episodic model, and $\tilde{O}\left(\frac{D_\infty^6|A|^4\log|\mathcal{W}|}{(1-\gamma)^{18}\alpha^4\varepsilon^6}\right)$ in the $\nu$-reset model.*

*Proof of Theorem 7.* The broad scheme here is to utilize an equivalence between Algorithm 1 and Algorithm 4 on the function $V^\pi$ (or $V_\nu^\pi$ in the $\nu$-reset model), to which Theorem 9 applies.

To this end, firstly, note $V^\pi$ is $\frac{1}{1-\gamma}$-bounded. Define a norm $\|\cdot\|_{\infty,1} : \mathbb{R}^{|S|\times|A|} \to \mathbb{R}$ as $\|x\|_{1,\infty} = \max_{s\in S}\sum_{a\in A}|x_{s,a}|$. Further, observe that for any policy $\pi : S \to \Delta_A$, $\|\pi\|_{\infty,1} = 1$. The following lemma specifies the smoothness of $V^\pi$ in this norm.

**Lemma 16.** $V^\pi$ is $\frac{2\gamma}{(1-\gamma)^3}$-smooth in the $\|\cdot\|_{\infty,1}$ norm.

To be able to interpret Algorithm 1 as an instantiation of the algorithmic template Algorithm 4 presents, we need to show that $\pi_t'$ (Line 3-10) serves as an approximate linear optimizer for $\nabla V^{\pi_{t-1}}$. This will imply that the iterates produced by the two algorithms coincide. Indeed, Claim 17 demonstrates that $\pi_t'$ serves a linear optimizer over gradients of the function $V^\pi$; the suboptimality specifies $\epsilon_0$.

**Claim 17.** *Let $\beta = \sqrt{\frac{1}{\alpha N}}$, and $\eta_{2,n} = \min\{\frac{2}{n}, 1\}$. Then, for any $t$, $\pi_t'$ produced by Algorithm 1 satisfies with probability $1 - \delta$*

$$\max_{\pi\in\Pi}(\nabla V_\mu^{\pi_{t-1}})^\top(\pi - \pi_t') \;\leq\; \frac{2|A|}{(1-\gamma)^2\alpha}\left(\frac{2}{\sqrt{N}} + \varepsilon_W\right)$$

Finally, observe that it is by construction that $\pi_t \in \sqcap$. Therefore, in terms of the previous section, $\mathcal{K}$ is the class of all policies, $\mathcal{K}_1 = \sqcap$, $\mathcal{K}_2 = \Pi$.

In the episodic model, we wish to invoke the second part of Theorem 9. The next lemma establishes gradient-domination properties of $V^\pi$ to support this.

**Lemma 18.** $V^\pi$ is $\left(C_\infty, \frac{1}{1-\gamma}C_\infty\mathcal{E}(\Pi,\mathbb{N}),\mathbb{N},\Pi\right)$-gradient dominated, i.e. for any $\pi \in \mathbb{N}$:

$$V^* - V^\pi \le C_\infty\left(\frac{1}{1-\gamma}\mathcal{E}(\Pi,\mathbb{N}) + \max_{\pi'\in\Pi}(\nabla V^\pi)^\top(\pi'-\pi)\right)$$

Deriving $\kappa,\tau$ from the above lemma along with $\epsilon_0$ from Claim 17, as a consequence of the second part of Theorem 9, we have with probability $1 - NT\delta$

$$V^* - V^{\bar\pi} \le C_\infty\frac{\mathcal{E}(\mathbb{N},\Pi)}{1-\gamma} + \frac{4C_\infty^2}{(1-\gamma)^3 T} + \frac{4|A|C_\infty}{(1-\gamma)^2\alpha\sqrt{N}}$$
$$+ \frac{2|A|C_\infty}{(1-\gamma)^2\alpha}\varepsilon_W.$$

Similarly, in the $\nu$-reset model, the first part of Theorem 9 provides a local-optimality guarantee for $V_\nu^\pi$. Lemma 19 provides a bound on the function-value gap (on $V^\pi$) provided such local-optimality conditions.

**Lemma 19.** For any $\pi \in \mathbb{N}$, we have

$$V^* - V^\pi \le \frac{1}{1-\gamma}D_\infty\left(\frac{1}{1-\gamma}\mathcal{E}_\nu(\Pi,\mathbb{N}) + \max_{\pi'\in\Pi}(\nabla V_\nu^\pi)^\top(\pi'-\pi)\right).$$

Again, using the bound on $\max_{\pi'\in\Pi}(\nabla V_\nu^{\bar\pi})^\top(\pi'-\bar\pi)$ Theorem 9 provides, we have that with probability $1 - 2NT\delta$

$$V^* - V^{\bar\pi} \le \frac{D_\infty\mathcal{E}_\nu(\mathbb{N},\Pi)}{(1-\gamma)^2} + \frac{2D_\infty}{(1-\gamma)^3\sqrt{T}}$$
$$+ \frac{2|A|D_\infty}{(1-\gamma)^3\alpha}\left(\frac{2}{\sqrt{N}} + \varepsilon_W\right)$$
$$+ \frac{48|A|D_\infty}{(1-\gamma)^3\sqrt{P}}\log\frac{1}{\delta}$$

$\square$

# E  Analysis for Boosting with Weak Online Learning (Proof of Theorem 15)

*Proof of Theorem 15.* Similar to the proof of Theorem 7, we establish an equivalence between Algorithm 1 and Algorithm 4 on the function $V^\pi$ (or $V_\nu^\pi$ in the $\nu$-reset model), to which Theorem 9 applies provided smoothness (see Lemma 16).

Indeed, Claim 20 demonstrates $\pi_t'$ serves a linear optimizer over gradients of the function $V^\pi$, and provides a bound on $\epsilon_0$. As before, observe that it is by construction that $\pi_t \in \mathbb{N}$.

**Claim 20.** Let $\beta = \sqrt{\frac{1}{\alpha N}}$, and $\eta_{2,n} = \min\{\frac{2}{n},1\}$. Then, for any $t$, $\pi_t'$ produced by Algorithm 5 satisfies with probability $1 - \delta$

$$\max_{\pi\in\Pi}(\nabla V_\mu^{\pi_{t-1}})^\top(\pi - \pi_t') \le \frac{2|A|}{(1-\gamma)^2\alpha}\left(\frac{2}{\sqrt{N}} + \frac{R_W(M)}{M} + \sqrt{\frac{16\log\delta^{-1}}{M}}\right)$$

In the episodic model, one may combine the second part of Theorem 9, which provides a bound on function-value gap for gradient dominated functions, which Lemma 18 guarantees, to conclude with probability $1 - T\delta$

$$V^* - V^{\bar\pi} \le \frac{C_\infty\mathcal{E}(\mathbb{N},\Pi)}{1-\gamma} + \frac{4C_\infty^2(\mathbb{N})}{(1-\gamma)^3 T} + \frac{4|A|C_\infty}{(1-\gamma)^2\alpha\sqrt{N}}$$
$$+ \frac{2|A|C_\infty}{(1-\gamma)^2\alpha}\frac{R_W(M)}{M} + \frac{8|A|C_\infty\log\delta^{-1}}{(1-\gamma)^2\alpha\sqrt{M}}.$$

Similarly, in the $\nu$-reset model, Lemma 19 provides a bound on the function-value gap provided local-optimality conditions, which the first part of Theorem 9 provides for. Again, with probability $1 - T\delta$

$$V^* - V^{\bar{\pi}} \leq \frac{D_\infty \mathcal{E}_\nu(\mathbb{\Pi}, \Pi)}{(1-\gamma)^2} + \frac{2D_\infty}{(1-\gamma)^3} \left( \frac{1}{\sqrt{T}} + \frac{|A|}{\alpha} \left( \frac{2}{\sqrt{N}} + \frac{R_{\mathcal{W}}(M)}{M} + \frac{4\log \delta^{-1}}{\sqrt{M}} \right) + \frac{24|A|}{\sqrt{P}} \log \frac{1}{\delta} \right).$$

$\square$

# F  Proofs of Supporting Claims

## F.1  Guarantees on the sampling algorithm

*Proof of Lemma 11.* Recall $\nabla_\pi V^\pi$ denotes the gradient with respect to the $|S| \times |A|$-sized representation of the policy $\pi$ – the functional gradient. Then, using the policy gradient theorem [38, 35], it is given by,

$$\frac{\partial V_\mu^\pi}{\partial \pi(a|s)} = \frac{1}{1-\gamma} d_\mu^\pi(s) Q^\pi(s,a). \tag{1}$$

The following sources of randomness are at play in the sampling algorithm (Algorithm 3): the distribution $d^\pi$ (which encompasses the discount-factor-based random termination, the transition probability, and the stochasticity of $\pi$), and the uniform sampling over $A$. For a fixed $s, \pi$, denote by $\mathcal{Q}_s^\pi$ as the distribution over $\widehat{Q^\pi}(s, \cdot) \in \mathbb{R}^A$, induced by all the aforementioned randomness sources. To conclude the claim, observe that by construction

$$\mathbb{E}_{\mathcal{Q}^\pi(s,\cdot)}[\widehat{Q^\pi}(s,\cdot)|\pi, s] = Q^\pi(s, \cdot). \tag{2}$$

$\square$

## F.2  Non-convex Frank-Wolfe method (Theorem 9)

*Proof of Theorem 9.* **Non-convex general case.** Note that for any timestep $t$, it holds due to smoothness that

$$f(x_t) = f(x_{t-1} + \eta(z_t - x_{t-1})) \tag{3}$$

$$\geq f(x_{t-1}) + \eta \nabla_{t-1}^\top (z_t - x_{t-1}) - \eta^2 \frac{L}{2} D^2. \tag{4}$$

Let $t' = \arg\min_t f(x_t) - f(x_{t-1})$. Note that by telescoping over function-value differences across successive iterates, we get

$$f(x_{t'}) - f(x_{t'-1}) \leq \frac{1}{T} \Big( f(x_T) - f(x_0) \Big) \leq \frac{2H}{T}.$$

Combining with (4), and plugging in $\eta$, we get

$$\nabla_{t'-1}^\top (z_{t'} - x_{t'-1}) \leq \eta L D^2/2 + \frac{2H}{T\eta}$$

$$\leq \sqrt{\frac{2LD^2 H}{T}}.$$

To conclude the claim for the non-convex general case, observe that since $z_{t'} = \mathcal{O}(\nabla_{t'-1})$, it follows by the oracle definition that

$$\max_{u \in \mathcal{K}_2} \nabla_{t'-1}^\top u \leq \nabla_{t'-1}^\top z_{t'} + \epsilon_0.$$

**Gradient-dominated case.** Let $x^* = \arg\max_{x \in \mathcal{K}} f(x)$ and let $h_t = f(x^*) - f(x_t)$.

$$h_t \leq h_{t-1} - \eta_t \nabla_{t-1}^\top (z_t - x_{t-1}) + \eta_t^2 \frac{L}{2} D^2$$

(by smoothness)

$$\leq h_{t-1} - \eta_t \max_{y \in \mathcal{K}_2} \eta_t \nabla_{t-1}^\top (y - x_{t-1}) + \eta_t^2 \frac{L}{2} D^2 + \eta_t \epsilon_0$$

(by oracle guarantee)

$$\leq h_{t-1} - \frac{\eta_t}{\kappa} (f(x^*) - f(x_{t-1})) + \eta_t^2 \frac{L}{2} D^2 + \eta_t \left( \epsilon_0 + \frac{\tau}{\kappa} \right)$$

(by gradient domination)

$$= \left( 1 - \frac{\eta_t}{\kappa} \right) h_{t-1} + \eta_t^2 \frac{L}{2} D^2 + \eta_t \left( \epsilon_0 + \frac{\tau}{\kappa} \right).$$

The theorem then follows from the following claim.

**Claim 21.** *Let $C \geq 1$. Let $g_t$ be a H-bounded positive sequence such that*

$$g_t \leq \left( 1 - \frac{\sigma_t}{C} \right) g_{t-1} + \sigma_t^2 D + \sigma_t E.$$

*Then choosing $\sigma_t = \min\{1, \frac{2C}{t}\}$ implies $g_t \leq \frac{2C^2 \max\{2D, H\}}{t} + CE$.*

$\square$

### F.3 Smoothness of value function (Lemma 16)

*Proof of Lemma 16.* Consider any two policies $\pi, \pi'$. Using the Performance Difference Lemma (Lemma 3.2 in [2], e.g.) and Equation **??**, we have

$$|V^{\pi'} - V^\pi - \nabla V^\pi (\pi' - \pi)|$$

$$= \frac{1}{1-\gamma} \left| \mathbb{E}_{s \sim d^{\pi'}} \left[ Q^\pi(\cdot|s)^\top (\pi'(\cdot|s) - \pi(\cdot|s)) \right] \right.$$

$$\left. - \mathbb{E}_{s \sim d^\pi} \left[ Q^\pi(\cdot|s)^\top (\pi'(\cdot|s) - \pi(\cdot|s)) \right] \right|$$

$$\leq \frac{1}{(1-\gamma)^2} \|d^{\pi'} - d^\pi\|_1 \|\pi' - \pi\|_{\infty,1}.$$

The last inequality uses the fact that $\max_{s,a} Q^\pi(s,a) \leq \frac{1}{1-\gamma}$. It suffices to show $\|d^{\pi'} - d^\pi\|_1 \leq \frac{\gamma}{1-\gamma} \|\pi' - \pi\|_{\infty,1}$. To establish this, consider the Markov operator $P^\pi(s'|s) = \sum_{a \in A} P(s'|s, a)\pi(a|s)$ induced by a policy $\pi$ on MDP $M$. For any distribution $d$ supported on $S$, we have

$$\|(P^{\pi'} - P^\pi)d\|_1$$

$$= \sum_{s'} \left| \sum_{s,a} P(s'|s, a)d(s)(\pi'(a|s) - \pi(a|s)) \right|$$

$$\leq \sum_{s'} P(s'|s, a)\|d\|_1 \|\pi' - \pi\|_{\infty,1}$$

$$\leq \|\pi' - \pi\|_{\infty,1}.$$

Using sub-additivity of the $l_1$ norm and applying the above observation $t$ times, we have for any $t$

$$\|((P^{\pi'})^t - (P^\pi)^t)d\|_1 \leq t\|\pi' - \pi\|_{\infty,1}.$$

Finally, observe that

$$\|d^{\pi'} - d^{\pi}\|_1 \leq (1-\gamma) \sum_{t=0}^{\infty} \gamma^t \|((P^{\pi'})^t - (P^{\pi})^t) d_0\|_1$$

$$\leq \|\pi' - \pi\|_{\infty,1} (1-\gamma) \sum_{t=0}^{\infty} t\gamma^t$$

$$= \frac{\gamma}{1-\gamma} \|\pi' - \pi\|_{\infty,1}.$$

$\square$

### F.4  Gradient domination (Lemma 18 and Lemma 19)

*Proof of Lemma 18.* Invoking Lemma 4.1 from [2] with $\mu = d_0$, we have

$$V^* - V^{\pi} \leq \left\| \frac{d^{\pi^*}}{d^{\pi}} \right\|_{\infty} \max_{\pi_0} (\nabla V^{\pi})^{\top} (\pi_0 - \pi)$$

$$\leq C_{\infty} (\max_{\pi_0} (\nabla V^{\pi})^{\top} \pi_0 - \max_{\pi' \in \Pi} (\nabla V^{\pi})^{\top} \pi'$$

$$+ \max_{\pi' \in \Pi} (\nabla V^{\pi})^{\top} (\pi' - \pi)).$$

Finally, with the aid of Equation **??**, observe that

$$\max_{\pi_0} (\nabla V^{\pi})^{\top} \pi_0 - \max_{\pi' \in \Pi} (\nabla V^{\pi})^{\top} \pi'$$

$$= \min_{\pi' \in \Pi} \frac{1}{1-\gamma} \mathbb{E}_{s \sim d^{\pi}} \left[ \max_a Q^{\pi}(s,a) - Q^{\pi}(\cdot|s)^{\top} \pi' \right]$$

$$\leq \frac{1}{1-\gamma} \mathcal{E}(\Pi, \mathbb{\Pi}).$$

$\square$

*Proof of Lemma 19.* Invoking Lemma 4.1 from [2] with $\mu = \nu$, we have

$$V^* - V^{\pi}$$

$$\leq \frac{1}{1-\gamma} \left\| \frac{d^{\pi^*}}{\nu} \right\|_{\infty} \max_{\pi_0} (\nabla V_{\nu}^{\pi})^{\top} (\pi_0 - \pi)$$

$$\leq \frac{1}{1-\gamma} D_{\infty} (\max_{\pi_0} (\nabla V_{\nu}^{\pi})^{\top} \pi_0 - \max_{\pi' \in \Pi} (\nabla V_{\nu}^{\pi})^{\top} \pi'$$

$$+ \max_{\pi' \in \Pi} (\nabla V_{\nu}^{\pi})^{\top} (\pi' - \pi)).$$

Again, with the aid of Equation **??**, observe that

$$\max_{\pi_0} (\nabla V_{\nu}^{\pi})^{\top} \pi_0 - \max_{\pi' \in \Pi} (\nabla V_{\nu}^{\pi})^{\top} \pi'$$

$$= \min_{\pi' \in \Pi} \frac{1}{1-\gamma} \mathbb{E}_{s \sim d_{\nu}^{\pi}} \left[ \max_a Q^{\pi}(s,a) - Q^{\pi}(\cdot|s)^{\top} \pi' \right]$$

$$\leq \frac{1}{1-\gamma} \mathcal{E}_{\nu}(\Pi, \mathbb{\Pi}).$$

$\square$

### F.5  Supervised linear optimization guarantees

*Proof of Claim 10.* The internal boosting subroutine of Algorithm 1, that is presented in Algorithm 5, is an instantiation of Algorithm 3 from [19], specializing the decision set to be $\Delta_A$. To note the equivalence, note that in [19] the algorithm is stated assuming that the center-of-mass of the decision

set is at the origin (after a coordinate transform); correspondingly, the update rule in Algorithm 1 can be written as

$$(\tilde{\pi}_n - \pi) = (1 - \eta_{2,n})(\tilde{\pi}_{n-1} - \pi) + \frac{\eta_{2,n}}{\alpha}(\mathcal{A}_{t,n} - \pi).$$

For any state $s$, $\pi(\cdot|s) = \frac{1}{A}\mathbf{1}_{|A|}$ corresponds to the center-of-mass of $\Delta_A$. Finally, note that maximizing $f^\top x$ over $x \in \mathcal{K}$ is equivalent to minimizing $(-f)^\top x$ over the same domain. Therefore, we can apply previous result on boosting for statistical learning from [19] (Theorem 13). Note that $\widehat{Q^\pi}(s, \cdot)$ produced by Algorithm 3 satisfies $\|\widehat{Q^\pi}(s, \cdot)\| = \frac{|A|}{1-\gamma}$. Let $\mathcal{D}_t$ be the distribution induced by the trajectory sampler in round $t$. This yields the bound in the claim. $\qquad\square$

*Proof of Claim 17.* Lemma 11 allows us to restate the guarantees from Claim 10 in terms of linear optimization over functional gradients. The conclusion thus follows immediately by combining Lemma 11 and Theorem 10. $\qquad\square$

### F.6 Online linear optimization guarantees (Claim 20)

*Proof of Claim 20.* In a similar vein to the proof of Claim 17, here we state the a result on boosting for online convex optimization (OCO) from [19] (Theorem 6), the counterpart of Theorem **??** for the online weak learning case.

**Theorem 22.** *Let $\beta = \sqrt{\frac{1}{\alpha N}}$, and $\eta_{2,n} = \min\{\frac{2}{n}, 1\}$. Then, for any $t$, $\Gamma[\tilde{\pi}_{t,m,N}]$ produced by Algorithm 5 satisfies*

$$\max_{\pi \in \Pi} \sum_{m=1}^{M} \left[ \hat{Q}_{t,m}^\top \pi(s_{t,m}) \right] - \sum_{m=1}^{M} \left[ \hat{Q}_{t,m}^\top \Gamma[\tilde{\pi}_{m,N}](s_{t,m}) \right] \leq \frac{2|A|}{(1-\gamma)\alpha} \left( \frac{2M}{\sqrt{N}} + R_{\mathcal{W}}(M) \right).$$

Next we invoke online-to-batch conversions. Note that in Algorithm 5, $(s_{t,m}, \hat{Q}_{t,m})$ for any fixed $t$ is sampled i.i.d. from the same distribution. Therefore, we can apply online-to-batch results, i.e. Theorem 9.5 in [17], on Theorem 22 to get

$$\max_{\pi \in \Pi} \mathbb{E}_{(s,Q) \sim \mathcal{D}_t} \left[ Q^\top \pi(s) \right] - \mathbb{E}_{(s,Q) \sim \mathcal{D}_t} \left[ Q^\top \pi_t'(s) \right] \leq \frac{2|A|}{(1-\gamma)\alpha} \left( \frac{2}{\sqrt{N}} + \frac{R_{\mathcal{W}}(M)}{M} + \sqrt{\frac{16 \log \delta^{-1}}{M}} \right).$$

We finally invoke Lemma 11. $\qquad\square$

### F.7 Remaining proofs (Claim 21)

*Proof of Claim 21.* Let $T^* = \arg\max_t\{t : t \leq 2C\}$. For any $t \leq T^*$, we have $\sigma_t = 1$ and $g_t \leq H \leq \frac{2C^2 H}{t}$. For $t \geq T^*$, we proceed by induction. The base case ($t = T^*$) is true by the previous display. Now, assume $g_{t-1} \leq \frac{2C^2 \max\{2D, H\}}{t-1} + CE$ for some $t > T^*$.

$$\begin{aligned} g_t &\leq \left(1 - \frac{2}{t}\right) \left( \frac{2C^2 \max\{2D, H\}}{t-1} + CE \right) \\ &\quad + \frac{4C^2 D}{t^2} + \frac{2CE}{t} \\ &\leq CE + 2C^2 \max\{2D, H\} \left( \frac{1}{t-1} \left(1 - \frac{2}{t}\right) + \frac{1}{t^2} \right) \\ &= CE + 2C^2 \max\{2D, H\} \frac{t^2 - 2t + t - 1}{t^2(t-1)} \\ &\leq CE + 2C^2 \max\{2D, H\} \frac{t(t-1)}{t^2(t-1)}. \end{aligned}$$

$\qquad\square$