# OpenReview forum: "A Boosting Approach to Reinforcement Learning"
_NeurIPS.cc/2022/Conference — NeurIPS 2022 Accept_

### Official Review · Reviewer_TxBq · 2022-07-10

**Rating:** 6
**Confidence:** 3
**Soundness:** 3 good
**Presentation:** 2 fair
**Contribution:** 2 fair

**Summary:**

This paper proposes a boosting approach for Reinforcement learning. The authors provide sample and Oracle complexity guarantees for finding a near-optimal policy when a weak learner is available. The proposed method essentially implements the Frank-Wolfe (FW) algorithm where the objective is the value function $\pi \mapsto V^\pi$ which is non-convex. However, thanks to a gradient dominance property of the value function, the FW algorithm is guaranteed to converge to a good solution. The weak learners are used within an internal boosting procedure to approximately (and efficiently) implement the FW procedure.

**Questions:**

Is the following true: The idea of using FW algorithm in the context of RL was already explored by e.g. Agarwal et al, 2019, 2020. The main contribution of this paper is the efficient implementation of FW that leverages weak-learner?

**Limitations:**

The paper does not seem to be in a polished state yet. There are many typos that made it hard to follow certain points. Here, I list some of them (some are minor):
- Line 87: distrbution -> distribution
- Display just before line 124: there should not be a condition on \pi and s in the definition of V^\pi
- Definition 2: should $\ell$ be a 'gain' rather than a loss (you want to minimize losses and maximize gains)
- Line 156: there is a reference to a policy tree (in math symbol), which is not defined yet.
- Definition 3: seems incomplete. Are you using a particular projection map $\Gamma$? The text before the definition suggests that you going to give the full details of the projection map.
- Line 185: missing Lemma number
- Line 1 of Alg 1. You say that the parameter $P$ is going to be passed to the internal boost (alg 2). I do not see any $P$ using within alg 2. Besides, I would recommend using another letter and reserving $P$ for the transition probability.
- Line 4 of alg 1: $\pi_t$ -> $\pi_{t-1}$.
- Alg 1 and Alg 2: list $\eta_{1,t}$ and $\eta_{2,t}$ as parameters of the algorithm.
- line 207: the statement 'learning complexity $|\mathcal{W}|$' is not precise (you did not say what it means to have a certain learning complexity)
- Alg 2. You use both $D'_n$ and $D_{t,n}'$. Be consistent.
- Line 249. You use both $\mathcal{K}_2$ and $\mathcal{K}$. Use just the former.
- The second inequality in Theorem 9 (i.e. the RHS inequality just before line 251) should go after line 252. Currently, it suggests that Alg 4 achieves this guarantee without the gradient domination property.
- It should be Algorithm 2 in Claim 10.
- Line 276: testing -> test.

**Strengths And Weaknesses:**

The high-level idea behind the proposed method is elegant. This idea consists of using weak learners and a boosting procedure to efficiently implement an approximate linear optimization oracle over the space of policy and then using it to run the FW algorithm to optimize the value function. Although the idea of using the FW algorithm in the context of RL does not seem to be new,

However, the presentation could be improved in my opinion. For example, the workings of the internal boosting procedure Algorithm 2 are never really explained in the main body. Given that this is a major component of the proposed method, explanations of the step of the algorithm should be included in the main body. A high-level sketch of Claim 10 would also be good.

---

> ### Author Response · Authors · 2022-08-02
> **Author response**
>
> We thank the reviewer for their careful reading of our paper, and will incorporate the suggestions made in the final version.
>
> We remark that the use of FW to overcome the non-convexity of the value function in the policy space is indeed a central contribution here, even while disregarding the boosting setting (i.e. when strong supervised/online learners are available). This is reflected in the improved results as displayed in Table 2. Further, it is central to the results that hold for the boosting setting.
>
> Agarwal-Henaff-etal and Agarwal-Kakade-etal analyze the policy gradient method (and natural policy gradient variants), as opposed to Frank Wolfe.
>
> Agarwal-Brukhim-etal use Frank Wolfe for control of continuous dynamical systems. But they do so under the assumption that the cost associated with any policy (namely, the value function) is convex in the policy space – an assumption that does not hold for (even tabular) MDPs.
>
> Hazan-Kakade-etal use FW to solve for non-scalar-reward functionals. However, they operate over convex costs, and use a standard RL agent as a blackbox oracle; the latter is an output of our algorithm.
>
> In light of this clarification, we would like to ask the reviewer to possibly raise their score if they deem it appropriate.

---

### Official Review · Reviewer_2Czt · 2022-07-11

**Rating:** 8
**Confidence:** 4
**Soundness:** 4 excellent
**Presentation:** 4 excellent
**Contribution:** 3 good

**Summary:**

This paper introduces the idea of boosting for reinforcement learning.
It presenting a boosting algorithm which outputs a policy by aggregating the policies suggested by a number of weak learners.
The sample complexity of the algorithm is analyzed theoretically. Effect of boosting is established both in theory, and with experiments.

**Questions:**

Main questions/comments:

1) It is quite interesting to see how the sample complexity of the weak learner $m(\epsilon, \delta)$ translates to the one for the boosted learner. In particular, the new dependancy on $\vert A \vert^4$ in intriguing. Do you have any intuition on why the sudden dependancy on the size of the action set? Why does this only happen when only $\alpha <1$?

2) In general, would be interesting to see the $V^* - V^\pi$ bound, where for a given $N$, $M$, $T$, and $\gamma$, the error is worked out. A discussion on the terms of the right hand side of this bound (e.g. which results from the external boost, which comes from internal boost, and which is the oracle's error) would be insightful.

3) How do the rates in Thm 7 and Cor 8 in general compare with sample complexity of other RL methods? (Only the dependence on $\epsilon$ is compared.)

4) I could not understand the rule for for calculating a base policy $\mathcal A_n$. Can you explain the intuition behind constructing the new dataset? In particular what is $f_i$ and what do the terms in definition of $y_i$ present?

5) Regarding the online weak learner: Is this necessary to consider a deterministic definition instead of having it hold with $1-\delta$?
Can you motivate one choice of assumption over the other? e.g. when the online oracle assumption makes more sense than the supervised learner. Generally, would have been nice if more of Appendix C was included in the main text. And to see how the resulting sample complexities are compare.

6) Would be cool to read the opinion of the authors on which assumptions are artefacts of the analysis, and which can not be lifted. In particular, what would gain/lose by altering aspects of the oracle weak learner assumption.

7) Assume there was computational limitations, and increasing $N$ the number of boosted weak learners would come at the cost of decreasing $M$. I am curious to know how these should be set in practice. Or, if there's a setup that would be generally more favorable? Judging by the bound I'm guessing we would benefit more from having many weak learners that are trained on smaller datasets?

8) How important is $\gamma$ in practice? Do we benefit a lot from larger $\gamma$s?

9) It seems to me that the reported experiment results are for single runs. For the plots to be meaningful, you should run the experiments with different random seeds and present the curves with the standard error. Although, without a doubt, boosting deems effective.

Typos/small mistakes/notation issues:
- [Line 25]: I think "direct methods" is supposed to be emphasized rather than "so-called"?
- [Line 57-58]: I think some part of the sentence is accidentally erased.
- [Line 124]: This is slightly confusing, since the randomness is induced by both $\pi$ and $P$.
- [Line 151-152]: In the left hand side it should say $\ell(\mathcal{W}(D_m)(s))$. The $(s)$ is missing.
- [Line 156]: It should be $\pi \in \Pi$.
- [Line 158]: Again I think it should be $\Pi$.
- [Line 248]: There's a space or \n missing before "Oracle".
- [Line 183-186]: I couldn't understand this part. Rewording would be nice.
- [Algorithm 1]: $\eta_{1,t}$ should also be listed as an input to the algorithm.
- [Algorithm 1]: This is the first time that $M$ is used for the sample size of the weak learner. Better to be defined.
- [Algorithm 2]: The notation $D_n$ used for the dataset of size $M$ of the $n$-th weak learner can be confused with notation $D_m$ in Def 2, which refers to a dataset of size $m$.
- [Algorithm 2]: Line 5 should be $i \in [M]$.
- [Algorithm 2]: Not all of the input to the algorithm (e.g. $\alpha$ and $\eta_{2,n}$) are listed.
- [Algorithm 3]: Again the input to the algorithm are not listed. In particular $\gamma$.
- [Line 204-206]: $\mathcal E$ is used for the first time. I'm assuming $\mathcal E := \mathcal E_{d_0}$?
- There's small issue with notation. The transition matrix $P$ of the MDP has the same notation as $P$ the number of rollouts/episodes.


**Limitations:**

Reading the paper, I can not tell how realistic is the weak oracle assumption for an $\alpha$ that is sufficiently close to $1$. Specially in the case of the online weak learner. It is explained that this is a common assumption in the boosting literature. It is hard for me to see how this translates to when the dataset $D_m$ is sampled from a non-i.i.d. distribution (basically it is sampled via Alg 3).

**Strengths And Weaknesses:**

**Originality/Significance.** I found the idea to be quite cool, can certainly imagine how weak learners can be practical in sequential settings. In general, I think revisiting basic techniques from supervised learning is a helpful practice for advancing the field and understanding certain limitations of it. I learned quite a bit while reviewing the paper and it immediately had me thinking about other areas in sequential learning which the idea of using weak learners would be useful. I think this paper can potentially inspire many follow up ideas.

I am not thoroughly aware of this area of research and thus not in a place to comment about the contributions of the paper. This being said, and judging by the related works section, it seems to me that this paper is the first of its kind. Although, it seems that boosting was recently considered in the context of online learning.

**Clarity/Quality.** The paper is written really well. It was easy to follow, while this is not my immediate area of research.
I found the given background to be exactly enough. The contributions and the challenges were stated quite clearly. I couldn't understand Tables 1 and 2 right away, but after reading the paper this too became clear. More intuition could be given on certain design choices and the details of the algorithms. The theorems also lack interpretations, in my opinion.

I should mention that I did not verify the correctness of the results. I only checked to see if the bounds behave the way I expect them with certain variables. Which they did.

---

> ### Author Response · Authors · 2022-08-02
> **Author response**
>
> We thank the reviewer for their editorial suggestions and careful reading of our work. We will incorporate the reviewer’s suggestions in the final version. We clarify a few issues below:
>
> 1. Dependence on |A|: Indeed, our results (Theorem 7) on boosting weak learners for RL show an increased dependence on both |A| and the discount factor for $\alpha<1$, when compared to RL via supervised learning (Corollary 8). This stems from the algorithmic changes (the internal boosting procedure) needed to overcome the fact that the weak learners are only approximately ($\alpha$) good. We believe that the particular dependency on |A| we state for $\alpha<1$ is improvable, and it would be an interesting direction for future work to tighten these bounds.
> 2. We have similar statements in the full paper on Line 480, 486, 489, 501 that show the required dependence. We will take further care to demarcate which terms originate from internal boost vs the outer loop.
> 3. For RL via supervised learning, we note that the dependence of our results on other factors (like horizon, C_infty) is the same as that of conservative policy iteration. For boosting, we are unable to offer a direct comparison since we are unaware of other provable RL algorithms that operate with weak learners.
> 4. To overcome the approximate nature of weak learners, the internal boosting procedure iteratively aggregates the hypotheses produced by the weak learners. To boost the accuracy across these rounds, the weak learner is fed a dataset that aims to emphasize the *residuals*, or the part of the original dataset that the current mixture of hypotheses is not good at predicting. $f_i$ is this residual, and $y_i$ is an intermediate step used for its construction.
> 5. We can tolerate regret bounds that hold with high probability. Online learners provide a stronger learning guarantee in that they accommodate adversarial or worst case inputs, while supervised learners require their datasets to be drawn IID from some fixed distribution. Correspondingly, supervised weak learners are easier computationally and statistically to construct than online weak learners. By utilizing the stronger guarantees of an online weak learner, we are able to use a fewer number of these to construct a RL algorithm (as Table 1 shows). There is a trade-off between the weak learning models in this sense.
> 6. We mention a relaxation of the online learning model in Remark 14 in the full paper, that may be of interest to the reviewer.
> 7. Our analysis reflects that ideally one would want to set $N=M$ (or very closely). However these being worst case bounds, in practice this question needs further exploration.
> 8. The discount factor ($\gamma$) measures how much one cares about long-term rewards. Generally, RL problems with higher $\gamma$ are more difficult; in the extreme case when $\gamma=0$, RL reduces to a contextual bandit problem.
> 9. We furnish the plots with confidence bands on the last page of the appendix.

---

> > ### Comment · Reviewer_2Czt · 2022-08-03
> > **Response to Author's Response**
> >
> > Thank you for your response.
> >
> > It clarified some uncertainties, and I have updated my review accordingly.

---

### Meta-Review · Area_Chair_WTw2 · 2022-08-23

**Recommendation:** Accept
**Confidence:** Less certain

**Metareview:**

This paper introduces a boosting approach for RL. RL is reduced to a supervised learning problem. The proposed method implements the Frank-Wolfe algorithm where the objective is the value function is non-convex. The weak learners are used approximately (and efficiently) solve the problem. The paper gives a sample complexity bound for finding a near-optimal policy using weak learning assumption.

This paper provides an interesting novel approach for RL that is certainly of interest to Neurips community and as such is a valuable contribution to the conference,

**Award:**

No

---

### Decision · Program_Chairs · 2022-09-14

Accept